# Incoherence between Systemic Hemodynamic and Microcirculatory Response to Fluid Challenge in Critically Ill Patients

**DOI:** 10.3390/jcm10030507

**Published:** 2021-02-01

**Authors:** Paolo De Santis, Chiara De Fazio, Federico Franchi, Ottavia Bond, Jean-Louis Vincent, Jacques Creteur, Fabio Silvio Taccone, Sabino Scolletta

**Affiliations:** 1Department of Intensive Care, Hôpital Erasme, Cliniques Universitaires de Bruxelles Erasme, Université Libre de Bruxelles, Route de Lennik, 808-1070 Brussels, Belgium; desantis.com@hotmail.com (P.D.S.); dfzchr@unife.it (C.D.F.); federico.franchi@dbm.unisi.it (F.F.); ottaviabond@gmail.com (O.B.); jlvincen@intensive.org (J.-L.V.); jcreteur@ulb.ac.be (J.C.); 2Department of Medicine, Surgery and Neuroscience, Emergency-Urgency and Organ Transplantation, University Hospital of Siena, 53100 Siena, Italy; sabino.scolletta@dbm.unisi.it

**Keywords:** fluid challenge, fluid responsiveness, microcirculation, tissue perfusion

## Abstract

Background: The aim of the study was to assess the coherence between systemic hemodynamic and microcirculatory response to a fluid challenge (FC) in critically ill patients. Methods: We prospectively collected data in patients requiring a FC whilst cardiac index (CI) and microcirculation were monitored. The sublingual microcirculation was assessed using the incident dark field (IDF) CytoCam device (Braedius Medical, Huizen, The Netherlands). The proportion of small perfused vessels (PPV) was calculated. Fluid responders were defined by at least a 10% increase in CI during FC. Responders according to changes in microcirculation were defined by at least 10% increase in PPV at the end of FC. Cohen’s kappa coefficient was measured to assess the agreement to categorize patients as “responders” to FC according to CI and PPV. Results: A total of 41 FC were performed in 38 patients, after a median time of 1 (0–1) days after ICU admission. Most of the fluid challenges (39/41, 95%) were performed using crystalloids and the median total amount of fluid was 500 (500–500) mL. The main reasons for fluid challenge were oliguria (*n* = 22) and hypotension (*n* = 10). After FC, CI significantly increased in 24 (58%) cases; a total of 19 (46%) FCs resulted in an increase in PPV. Both CI and PPV increased in 13 responders and neither in 11; the coefficient of agreement was only 0.21. We found no correlation between absolute changes in CI and PPV after fluid challenge. Conclusions: The results of this heterogenous population of critically ill patients suggest incoherence in fluid responsiveness between systemic and microvascular hemodynamics; larger cohort prospective studies with adequate a priori sample size calculations are needed to confirm these findings.

## 1. Introduction

Fluid therapy is the first line treatment in patients with acute circulatory failure, as fluids can increase cardiac output and improve tissue perfusion. Although fluid administration remains an early intervention in this setting, the optimal timing, the amount of fluids as well as the effectiveness of such intervention should always be carefully evaluated to reduce the risk of fluid overload [1]. Therefore, the hemodynamic assessment of fluid responsiveness using the fluid challenge approach is one of the most effective way to identify patients who can benefit from volume expansion, avoiding the risk of volume overload and systemic complications [2]. The hemodynamic response to a fluid challenge is defined according to the Frank–Starling principle. However, a practical limitation of this approach is that patients who do not respond to fluid administration would receive fluids, since the responsiveness to the fluid challenge test can be evaluated after fluid administration. Furthermore, changes in tissue perfusion are not correlated with changes in macrohemodynamics (i.e., cardiac index) in patients requiring fluid administration [3,4].

With the introduction of new microcirculatory imaging techniques, such as video-microscopy, a direct assessment of the microcirculation at bedside in order to directly evaluate tissue perfusion is now possible. Different studies have shown that fluid administration improved microvascular perfusion in patients with sepsis or trauma [3,5] together with an increase in cardiac index (CI); however, these effects were independent on global hemodynamic effects and of the type of solution, which may translate into a different “responsiveness status” of the macro and the microcirculation.

The main purposes of this study were therefore to compare the coherence of fluid responsiveness of the macro and microcirculation and to assess the relationship between changes in microcirculatory and macrohemodynamic variables.

## 2. Materials and Methods

This was a retrospective analysis including patients admitted to a 35-bed mixed medical-surgical intensive care unit (ICU). The study was approved by the hospital ethics committee (P2017/225), which waived the need for written informed consent because of the observational and retrospective nature of the study. All adult patients admitted between February and June 2016 to the Department of Intensive Care at Erasme Hospital, Brussels (Belgium) were eligible if: (a) underwent a fluid challenge, according to the decision of the attending physician; (b) had any form of invasive hemodynamic monitoring system that allowed the measurement of beat-to-beat CI; (c) underwent microcirculatory evaluation as routine monitoring of tissue perfusion [6]; (d) fluid challenge occurred during working hours. We excluded patients who received changes in the infusion speed of inotropes, sedation and ventilation mode between the two detection times.

The principal reason for inclusion was the clinical need for fluid therapy according to one or more predefined signs of impaired organ perfusion: mean arterial pressure (MAP) <60 mmHg or >40 mmHg below normal values for at least 15 min; tachycardia >100 bpm; oliguria <0.5 mL/kg per hour for at least 3 h; metabolic acidosis (base excesse, BE < −2 mEq/L) with hyperlactatemia (>2 mmol/L); reduced cardiac index (CI < 2.2 L/min·m^2^) associated with dynamic fluid responsiveness predicting an increase in cardiac output after fluid therapy. Patients enrolled could be in spontaneous breathing or ventilated in assisted or controlled mode. Additionally, inotropic/vasopressors or vasodilators agents could be administered. Both rhythmic and arrhythmic patients were included.

### 2.1. Data Collection

We collected demographic and anthropometric data. Treatments with mechanical ventilation, inotropes and vasoactive drugs, sedative-hypnotics and analgesics drugs were recorded. For a correct macrohemodynamic monitoring, presence and correct functioning of the catheter for invasive arterial pressure detection, of the central venous access and of the advanced monitoring device (EV1000 device, Edwards Lifesciences, Irvine, CA, USA; MostCare device, Vygon, Ecouen, France; PiCCO device, Getinge, Halmstad Municipality, Halland County, Sweden) were evaluated; some of these patients also had pulmonary artery catheter, as decision of the treating physician, which was not used to assess CI for this study. The following hemodynamic parameters were recorded at the baseline and after the fluid challenge: heart rate; mean arterial pressure; central venous pressure; CI; lactate blood level; central oxygen venous saturation (ScvO_2_) value; veno–arterial gap of carbon dioxide (vaCO_2_ gap).

For the microhemodynamic monitoring, incident dark field (CytoCam, Braedius Medical, Huizen, The Netherlands) was used. After an appropriate preparation of the oropharyngeal mucosa at the base of the tongue (removal of salivary and/or blood secretion, endotracheal tube custody in the intubated patient, procedure delineation and immobility request in the conscious patient), and avoiding pressure artefacts, images were obtained from five different locations within the sublingual region. The following parameters were calculated, using the De Backer score [7]: total small vessel density (TVD); proportion of perfused vessels (PPV); perfused microvascular density (PVD). Semiquantitative analysis of the microcirculatory flow was performed as previously described by Boerma et al. [8]. Each image was divided into four equal quadrants and for each one a quantification of flow was scored (no flow: 0; intermittent flow: 1; sluggish flow: 2; continuous flow: 3). The determination of microvascular flow index (MFI) was based on the predominant type of flow in four quadrants and averaged over the values obtained in each one. We also calculated the heterogeneity index, following the method of Trzeciak et al. [9], based on MFI and PPV values.

All the macro and micro hemodynamic parameters were collected at two different times: at baseline (T0), i.e., immediately before starting the fluid-challenge test, and at the end of the infusion (T1). No change in infusion speed of inotropes, sedation and ventilation mode was performed between the two detection times, as routine practice to assess the effectiveness of the fluid challenge.

### 2.2. Definitions

Fluid challenge was defined as a fluid bolus (500 mL crystalloid solution or colloid solution depending on the choice of the attending physician) infused within a short time, maximum 30 min. Fluid responders were defined as patients whose variations in CI (CI-responders) or PPV (PPV-responders) after fluid challenge were 10% higher than the baseline value. This threshold was decided according to the inter-observer maximum variability in PPV assessment [10]. Whenever available (i.e., controlled volume ventilation with adequate depth of sedation and no spontaneous breathing, no arrhythmias, tidal volume >8 mL/kg of ideal body weight), pulse pressure variation >13% was used to identify “fluid responder” before the fluid challenge. No changes in vasopressor or inotropic agents, as well as sedative drugs, was performed during the study period.

### 2.3. Statistical Analysis

Data were expressed as median and interquartile range (25th to 75th percentiles) as the Kolmogorov–Smirnov test showed a non-normal of distribution of continuous variables. The Wilcoxon test was used to compare repeated measurements. Regression analysis was used to test the relationship between microcirculatory and hemodynamic variables. The changes of the variables after fluid challenge were computed as absolute changes from baseline. Cohen’s kappa coefficient was measured to assess agreement to categorize patients as “responders” to fluid challenge using the CI-responder and PPV-responder criteria. Differences between responders and non-responders to fluid challenge were analysed using Mann Whitney U-test. The ability of the PPV to follow variations or trends of CI after a fluid challenge was assessed by analysing the correlation between the changes of the two variables, which were calculated by subtracting the first from the second measurement (T1–T0). After excluding all the pairs of measurements where at least one value was zero, we analysed the direction of change of CI and PPV to assess the percentage of concordance between the two variables [11,12].

For the prediction of an increase of PPV > 10%, as the number of events was small, only variables measured at baseline with a *p* values < 0.05 at the univariate analysis would have been considered in the multivariable logistic regression.

Statistical analysis was performed using PRISM version 8.0 (GraphPad Company, San Diego, CA, USA), and IBM^®^ SPSS^®^ Statistics software, version 22.0 (IBM, Armonk, NY, USA) for Macintosh. For all statistical tests, a *p* < 0.05 was taken to indicate significance.

## 3. Results

A total of 38 patients (male gender, *n* = 26; median age of 64 (51–69) years) were included over the study period. Demographic characteristics, comorbidities and reasons of ICU admission are reported in Table 1.

Overall, 41 fluid challenges were performed, as three patients received a second fluid challenges on another day. Median time from ICU admission to fluid challenge was 1 (0–1) days. Most of the fluid challenges (39/41 95%) were performed using crystalloids (Plasmalyte in 37, NaCl 0.9% and Ringer’s Lactate in 1 each), while two patients received 4% albumin. The median total amount of administered fluid and the time of the infusion were 500 (500–500) mL and 20 (15–20) minutes, respectively. The main reasons for fluid challenge were oliguria (*n* = 22), hypotension (*n* = 10), low cardiac output (*n* = 3) and suspected hypovolemia (*n* = 2).

At the time of the fluid challenge, 24 (59%) patients were on norepinephrine (median dose 0.21 (0.11–0.49) µg/kg/min), with 11 of these 24 patients receiving also dobutamine. In total, 24 (59%) patients were on continuous intravenous sedative (midazolam in 13 and propofol in 6) or analgesic (morphine in 6 and remifentanil in 18) therapy. Among the 16 patients on controlled mechanical ventilation, median pulse pressure variation was 14% (ranges: 5–21%), with 10/16 (62%) having criteria for fluid responsiveness prediction. After fluid challenge, heart rate and venous arterial CO_2_ gap significantly decreased; conversely, MAP, central venous pressure, CI, and central venous saturation significantly increased. In addition, TVD, PPV, and MFI significantly increased after fluid challenge, while the heterogeneity of MFI and PPV significantly decreased. Hemodynamic and microcirculatory parameters before and after the fluid challenge are shown in Table 2.

A total of 24 (58%) patients were considered as CI-responders after fluid challenge, while 19 (46%) patients were considered as PPV-responders. In total, 13 patients (31%) were “responders” and 11 (26%) were “non-responders” for both CI and PPV changes (Table 3). As such, incoherence between CI-responsiveness and PPV-responsiveness was observed for 18 (43%) fluid challenged. The coefficient of agreement between CI-responder and PPV-responder to the fluid challenge was 0.21. The proportion of responders to CI and PPV in the subgroup of patients treated with norepinephrine or sedative/analgesics drugs is reported in Appendix A.

We found no correlation between absolute changes in CI and PPV after fluid challenge (Figure 1); similar results were found when changes in CI and MFI were considered.

A significant but weak correlation was found between the absolute change in PPV after fluid challenge and the baseline values of PPV (R^2^ 0.260, *p* < 0.001), as well as absolute change in MFI after fluid challenge and the baseline values of MFI (R^2^ 0.436, *p* < 0.001) (Figure 1). Similar results were obtained in the subgroups analysis, according to the use of norepinephrine or sedative/analgesic drugs (Appendix A). To evaluate the concordance between CI and PPV or MFI, we excluded from the analysis the pairs of data when at least one variable between CI and PPV, or CI and MFI value was zero. The concordance of CI and PPV was 79% (31 of 39 pairs of data agreed) (Figure 1). Considering the variation of CI and MFI, the concordance of the two variables was 81% (30 of 37 pairs of data agreed) (Figure 1). In the univariate analysis, neither hemodynamic nor microcirculatory parameters at baseline were able to predict the response in PPV to the fluid challenge (Appendix A). Thus, no multivariable logistic regression was performed. Finally, among the 10 patients who were fluid-responder according to changes in pulse pressure before the fluid challenge, only seven of them were PPV-responder (70%). On the opposite, 2/6 patients (33%) who were “non-responder” based on the variations of pulse pressure before the fluid challenge were PPV-responders.

## 4. Discussion

In the present study, following a fluid challenge test, we observed a significant improvement in both macro and micro-hemodynamic parameters, including mean arterial blood pressure, central venous oxygen saturation, veno-arterial CO_2_ gap, microvascular density and flow. Importantly, in half of fluid challenges, we observed an incoherence between CI-responsiveness and PPV-responsiveness. There was no correlation between the absolute variations in CI and PPV after fluid challenge. A significant but weak correlation was found between absolute variations of PPV at the baseline and after fluid challenge, as previously reported [6,13].

Concordance between variations in macro and microhemodynamic parameters was weak; only 57% of the patients examined showed a significant improvement in CI (“macrocirculatory responsiveness”) and in terms of PPV (“microcirculatory responsiveness”) or absence of fluid responsiveness for both macro and micro-circulation. Coefficient of agreement was 0.21, indicating poor correlation between CI and PPV in the definition of “responder” or “non-responder” as a result of a fluid challenge. This fluid-responsiveness incoherence between macro and microcirculation was frequent (43% of fluid challenges) and provides a significant challenge for physicians. In clinical practice, CI-responsiveness is used to consider a “positive” response to fluid challenge, as increase in CI will eventually result in improved tissue perfusion. However, our study shows that increase in CI will not automatically translate in improved sublingual capillary density or flow in all patients; more importantly, some patients, who are considered as “non-responders” according to changes in CI, might still have significant improvement in microcirculatory flow. The concept of “fluid responsiveness” based on changed in CI might not be appropriate in several patients and changes in CI should not be the only tool to encourage or discourage volume loading, as changes in the microcirculation, which is the main determinant of tissue perfusion, are unpredictable. Interestingly, the physician may erroneously suspend the administration of fluids and prevent the patient to receive an effective intervention to restore microvascular perfusion when only CI-responsiveness is considered. As such, we considered that the concept of “hemodynamic incoherence”, i.e., a disorder in which resuscitation procedures aimed at the correction of systemic hemodynamic variables are ineffective in correcting microcirculatory perfusion [14], should be associated to “incoherence to fluid responsiveness”, where significant changes in macrohemodynamics cannot predict changes in microcirculatory variables.

In total, four types of loss of hemodynamic coherence have been proposed. In type 1, the main finding is the heterogeneous perfusion of the microvessels, which is typical in septic patients who have non-perfused vessels close to perfused capillaries. Type 2 is typically of hemodilution, which is responsible for the loss of RBC-filled capillaries and increased diffusion distance between RBCs and the tissue cells. In type 3, stasis of microcirculatory blood flow, due to increased systemic vascular resistance or central venous pressures, is observed. Finally, type 4 are observed in severe tissue edema, which is responsible for an increased distance between the capillaries and the cells and reduced oxygen delivery to tissues [14]. We could not really provide subgroups analyses on different types of hemodynamic incoherence to compare CI-responders and PPV-responders, because of the small cohort of included patients. Future studies should address whether this incoherence to fluid responsiveness is influenced by the pathophysiology of the microvascular dysfunction or could occurring independently ion the underlying disease.

In the literature, a number of studies have described such types of loss hemodynamic coherence [13,15,16,17]. In those studies, the resuscitation, resulting in normalization of systemic hemodynamic variables, did not result into an improvement of the microcirculation. This could probably explain why several studies, which targeted the “normalization” of systemic oxygen delivery variables, had negative results on patients’ outcome [18,19]. However, more importantly, this incoherence could also explain the increased morbidity and mortality related to these aggressive therapeutic strategies, for the risk related to aggressive interventions aimed at standardizing systemic parameters that will never cause the same improvement of the microcirculation. In brief, this management may not merely be futile, but even harmful. From this point of view, the correct evaluation of fluid resuscitation should not only include the improvement of systemic hemodynamic parameters, but also verify any improvement in the microcirculation.

Markers of organ perfusion such as lactate, peripheral temperature and capillary refill time can be used to identify loss of hemodynamic concordance between macro and microcirculation. However, such parameters have some limitations: lactates, for example, are a terminal metabolic product whose origin does not always reflect tissue hypoperfusion [20]. Peripheral temperature and refill time, on the other hand, are based on the integrity of the skin, whose function is primarily other than tissue oxygenation, such as thermoregulation; therefore, correlation with splanchnic and sublingual microcirculation may be deficient in the critical ill patient [21].

Our study presents several limitations. First, it is a monocentric and retrospective study with a small sample tested; moreover, no sample size calculation was performed and the heterogeneity of patients and reasons for fluid administration can significantly influence the power of the study and its conclusions. Second, we analyzed a heterogeneous population (i.e., different underlying diseases, use or vasopressors or inotropic agents) and the relationship between macro- and micro-hemodynamic is influenced by the underlying pathogenetic mechanism. Future larger cohort studies should specifically evaluate patients suffering from shock and multiple organ failure, in order to understand whether the severity of the disease might potentially affect the dynamic incoherence between macro and microcirculation. However, previous studies that only considered specific subgroups of patients, predominantly with sepsis [3,22], were not representative of the daily clinical reality in the ICU, often characterized by the presence of relatively mild hemodynamic alterations in a very heterogeneous population. Third, we performed only the IDF analysis of sublingual microcirculation: on the one hand, this is not always feasible and might limit its applicability in critically ill patients and, on the other hand, it is possible that the microcirculation in other organs may have different relationship with systemic hemodynamics after a fluid challenge [23]. Fourth, we analyzed “stabilized” patients and it is possible that the same analysis performed in the very early resuscitation phase of shock could have provided different results. In addition, the type of fluids used for the volume change could influence the macro- and micro-hemodynamic response in different ways. However, crystalloids have been used in most cases. Furthermore, the use of different types of fluids for the volume challenge represented a closer picture to daily reality. Fifth, we could not ensure an adequate calibration for some monitoring tools (i.e., EV1000, PiCCO) before the fluid challenge, as this may potentially influence the relative changes in cardiac output after therapeutic interventions. Sixth, since measurements were performed directly after completion of the fluid challenge, the results on macrohemodynamics and microcirculation are limited to the acute phase but cannot be extrapolated to assess the situation after a potential equilibration. Finally, we could not assess tissue oxygenation and it remains unknown whether changes in tissue perfusion would be associated with modification of metabolism (i.e., adaptive vs. pathological alterations).

## 5. Conclusions

In this heterogeneous patients’ population of critically ill patients, an incoherence between CI-responsiveness and PPV-responsiveness in almost half of fluid challenges was observed. Larger cohort prospective studies with adequate a priori sample size calculations are needed to confirm these studies. As such, the definition of fluid responsiveness may be challenged and include, in some patients, microcirculatory assessment to evaluate the effects of fluid administration on tissue perfusion. Whether this approach is feasible and might be used to titrate fluid administration in critically ill patients, has to be demonstrated in larger studies.

## Figures and Tables

**Figure 1 jcm-10-00507-f001:**
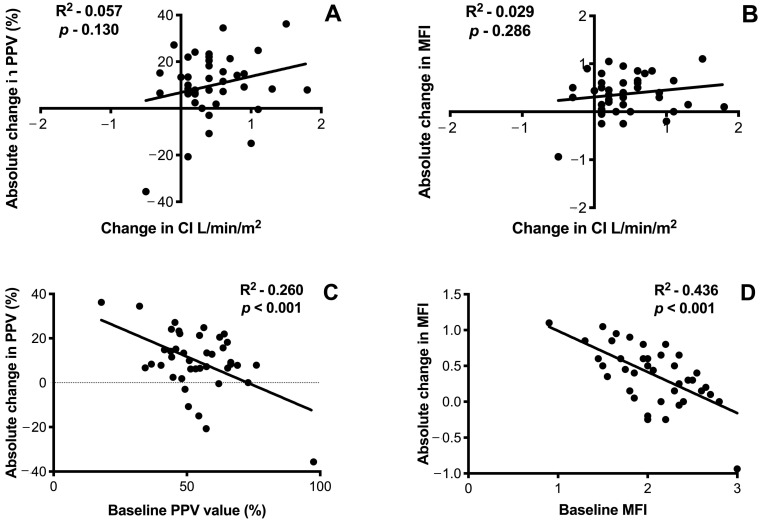
Correlation analysis. Upper panel: correlation between the absolute change in proportion of perfused small vessels (PPV) after fluid challenge and the absolute change of cardiac index (CI) (**A**), and between the absolute change in mean flow index (MFI) and the absolute change of CI (**B**) after fluid challenge. Lower panel: correlation between the absolute change in PPV after fluid challenge and the baseline values of PPV (**C**), and between the absolute change in MFI after fluid challenge and the baseline values of MFI (**D**).

**Table 1 jcm-10-00507-t001:** Demographics and data on the day of fluid challenge. Data are presented as median (IQRs) or count and percentage.

**Patients *n* = 38**	
Age, years	64 (51–69)
Height, cm	169 (160–180)
Weight, Kg	82 (73–100)
Body surface area, m^2^	1.91 (1.81–2.07)
Male, *n* (%)	26 (68)
APACHE II Score on admission	26 (8–42)
**Main reason of ICU admission**	
Cardiac arrest, *n* (%)	8 (21)
Post-cardiac surgery, *n* (%)	8 (21)
Cardiogenic shock, *n* (%)	6 (16)
Respiratory failure, *n* (%)	6 (16)
Hemorrhagic shock, *n* (%)	4 (11)
Liver transplantation, *n* (%)	3 (8)
Septic shock, *n* (%)	3 (8)
**Comorbid Diseases**	
COPD/Asthma, *n* (%)	4 (10)
Heart disease, *n* (%)	22 (58)
Diabetes, *n* (%)	3 (8)
Chronic renal failure, *n* (%)	4 (10)
Liver cirrhosis, *n* (%)	5 (13)
Previous neurological disease, *n* (%)	4 (10)
Immunosuppression, *n* (%)	4 (10)
**On the day of Fluid Challenge**	
Controlled ventilation, *n* (%)	16 (42)
Assisted ventilation, *n* (%)	13 (34)
Spontaneous breathing, *n* (%)	9 (24)
Sinus rhythm, *n* (%)	38 (100)
Sedation, *n* (%)	22 (58)
Norepinephrine, *n* (%)	22 (58)
Dobutamine, *n* (%)	11 (29)
**Reasons for Fluid challenge**	
Oliguria, *n* (%)	22 (58)
Hypotension, *n* (%)	10 (26)
Low cardiac output, *n* (%)	3 (8)
Suspected hypovolemia, *n*(%)	2 (5)
Crystalloids, *n* (%)	39 (95)
ICU mortality, *n* (%)	14 (37)

APACHE II Score, Acute Physiology and Chronic Health Evaluation II; COPD, chronic obstructive pulmonary disease; ICU, intensive care unit.

**Table 2 jcm-10-00507-t002:** Hemodynamic and microcirculatory parameters before and after the fluid challenge. Data were expressed as median (IQRs).

Variable	Pre-Fluid Challenge N = 41	Post-Fluid Challenge N = 41	*p* Value
Hemodynamics			
Heart rate, beat per minutes	92 (78–104)	84 (75–100)	<0.001
Mean arterial pressure, mmHg	70 (65–80)	75 (69–87)	0.001
Central venous pressure, mmHg	8 (6–8)	10 (8–14)	<0.001
Cardiac index, L/min/m^2^	2.6 (2.1–3.1)	3.0 (2.5–6.5)	<0.001
Lactate, mmol/L	1.7 (1.0–2.9)	1.8 (1.0–2.7)	0.181
Central oxygen venous saturation, %	70 (63–76)	72 (66–79)	0.002
Veno-arterial CO_2_ gap, mmHg	8 (6–10)	7 (6–10)	0.049
*Microcirculation*			
Total vessel density (TVD), mm/mm^2^	14.7 (13.2–18.2)	16.7 (14.8–18.7)	0.009
Density of perfused small vessel, mm/mm^2^	8.1 (5.9–10.3)	10.6 (8.0–12.5)	<0.001
Proportion of perfused small vessels (PPV), %	53 (45–63)	62 (55–74)	<0.001
Microvascular flow index, (MFI)	2.1 (1.8–2.4)	2.5 (2.1–2.8)	<0.001
Heterogeneity of MFI, %	0.47 (0.31–0.64)	0.34 (0.18–0.50)	0.002
Heterogeneity of PPV, %	0.60 (0.43–0.83)	0.43 (0.27–0.67)	0.009

PPV: proportion of perfused vessels; MFI: mean flow index.

**Table 3 jcm-10-00507-t003:** Proportion of patients responding to fluid challenge according to changes in cardiac index (CI, i.e., >10%) or to changes in the proportion of perfused small vessels (PPV, >10%). Total of fluid challenge = 41.

	Non Responders PPV	Responders PPV
Non responders CI	11	6
Responders CI	11	13

## Data Availability

Data are available on request to the authors.

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
