# Peer review of "Incoherence between Systemic Hemodynamic and Microcirculatory Response to Fluid Challenge in Critically Ill Patients"

_jcm, 2021, doi:10.3390/jcm10030507_

Round 1

Reviewer 1 Report

The study of De Santis et al. assessed the coherence between macrohemodynamics derived by pulse-contour analysis and sublingual microcirculatory parameters following a fluid challenge showing incoherence between macro- and microcirculatory parameters.

The advantages of this study are the inclusion of a heterogeneous intensive care collective and the use of sublingual microcirculatory evaluations based on current recommendations. The disadvantages would by the small number of patients for inclusion of heterogenic patients and heterogenic reasons for fluid administration as well as the retrospective analysis. Although the reviewer appreciates the aim of authors for performing a study on hemodynamic coherence in a mixed ICU-population during fluid challenge, there are several aspects that need to be addressed before the manuscript could be considered for publication.

Major aspects:

First of all, the language includes several spelling and grammar errors and should be corrected either by the authors (if possible) or by the help of native speakers or language editing services.

Second, there is concern about the ethical practice used by the authors. Although the authors have mentioned the retrospective character of the study, the prospective data collection as well as evaluation of hemodynamic monitoring systems prior data collection raises the question, whether this study was not in fact prospectively performed without consent of patients and the only reason for retrospective data analysis (which is not much different than an off-line data analysis) would be the speculation on a waiver from the ethics committee for the need of patient consent when emphasizing the "retrospective character". In this regard, microcirculatory evaluation would not be assumed to be part of routine evaluations but rather present study measurements. Although the reviewer is not familiar with local practice of the ethics committee, this issue needs to be clarified. If in fact this study was planned as retrospective analysis, at least this needs to be included as limitation.

Third, the references are incorrect, for example reference 13 and 17 are the same, and also 22 and 25 are the same. This needs to be corrected and the lack of reasonable care for this section may raise doubt about correct and accurate data collection and analysis for the entire manuscript.

Fourth, the authors should comment on how the number of patients included was derived? Is there a sample size calculation? The heterogeneity of patients and reasons for fluid administration raises doubt about the power of the study and whether the conclusions of the study could be based on this small sample size.

Finally, the conclusion in the abstract is limited to septic patients, however only a small amount of patients were septic. This should be corrected.

Minor aspects:
There are several minor aspects that should be commented on:

A major part of patients received fluids for being oliguric. Are these patients different from hypotensive patients or patients with low cardiac output? Did the urine output increase after fluid administration. Was there a difference in regard to micro- vs. macrocirculatory responders?

Introduction: The authors state that in septic and trauma patients fluids improved both microvascular perfusion AND cardiac output but that this was independent of global hemodynamic effects, what is meant by global hemodynamic effects? In regard to reference no. 3, in this study there was indeed a lack of microcirculatory improvement while cardiac output did improve in the later phase of sepsis. This should be corrected.

Methods:
1. Which "static variables" were used?

2. When using pulse pressure variations, what was the tidal volume used for these patients? This could possibly have influence prognostic value of pulse pressure variation in this study. Were all patients in which pulse pressure variations was measured under mandatory ventilation and under anesthesia of adequate depth, excluding spontaneous breathing efforts effecting the measurements?

3. Different methods for cardiac output evaluation were used, were these methods calibrated prior measurement and if so, how? This is especially important when using small defined values for cardiac output increases of only 10%.

4. Why was the fluid challenge not corrected for body weight instead of giving all patients the same amount? This may lead to errors when categorizing responders from non-responders, especially when using small values for cardiac output increases of only 10%.

5. Was there any time between completion of the fluid challenge and measurements to allow equilibration of macrohemodynamics as well as microcriculation after fluid challenge?

6. Why did the authors chose to report interquartile range instead of confidence intervals? In this regard, the reviewer had differences to assess, whether the data was normally distributed or not...

Results:

Are there any data on organ function (urine Output, FiO2/PaO2 ratio, liver function, neurologic function) that was associated with either macro- and/or microcirculatory parameters? What is the impact of sublingual microcirculation for this patient collective, especially since dissociation between sublingual and visceral microcirculation has already been shown. Since 58 % of patients receive fluids for correction of oliguria, impact on renal function for each type of "responder" would be of interest.

Limitation: Inclusion of patients both with and without inotropes and vasopressors should be included. How long did the patients receive inotropic support at time of study intervention, where there already in a stable condition or was vasopressor/inotropic support recently started?

Tables:

Table 1: Percentage and numbers of patients receiving dobutamine should be included.

Reviewer 2 Report

I have carefully revised the manuscript entitled "Incoherence between systemic hemodynamics and microcirculation response to fluid challenge in critically ill patients". The authors aimed at exploring the hemodynamic coherence in response to a fluid challenge in stabilized critically ill patients. The design of the study is clear, and the authors performed a single-center retrospective analysis of prospective observational data. According to the results, the authors concluded that there was an incoherence between CI-responsiveness and PPV-responsiveness in almost half of the patients included.

This study is interesting, and explores the hemodynamic coherence between macro and micro variables, and how it can change our classic Frank-Starling approach to fluid responsiveness. The methodology of the study is clear and properly explained. However, I might expose some concerns:

Minor comments:

1.- Patient selection and reason to trigger fluid challenge: According to the paper, the main reasons for receiving a fluid challenge were Oliguria, Hypotension, Low cardiac output and Suspected hypovolemia. From these 4 reasons, only Hypotension is considered a clear sign of shock that therefore should trigger an hemodynamic intervention such as a fluid challenge. Regarding the remaining 3 reasons: Oliguria can have multiple other causes other than hypoperfusion in critically ill patients. Low cardiac output per se should never trigger an hemodynamic intervention unless signs of tissue hypoperfusion are also present. Suspected hypovolemia is a vague concept, and also should never trigger a fluid challenge unless again signs of tissue hypoperfusion are present.

Therefore, the indication of a fluid challenge in a population of critically ill patients who are already stabilized (study patients at inclusion had MAP>65mmHg, normal lactate and normal SvcO2) from a cardiovascular point of view is debatable. In that regard, it would be particularly interesting to explore the hemodynamic coherence of patients who are in shock state (defined by presence of tissue hypoperfusion markers), since that is the target patient who is going to potentially benefit from fluid resuscitation.

2.- Central venous pressure: It would be interesting to see if changes in CVP were related to the micro (PPV/MFI) response after a fluid challenge. CVP has been proposed as the backpressure of the microcirculation, and therefore it might influence the microcirculatory response to fluids.

3.- Study population: 38 patients were included. In table 1, the main reason of ICU admission is only provided for 31 subjects (cardiac arrest 8, post-cardiac surgery 7, cardiogenic shock 4, respiratory failure 3, hemorrhagic shock 3, liver transplantation 3 and septic shock 3). Furthermore, it seems that 50% of the patient population were patients with acute cardiac injury (cardiac arrest, post-cardiac surgery and cardiogenic shock). I suggest this issue to be addressed in the discussion, as the underlying disease might influence the loss of hemodynamic coherence.

4.- Discussion: line 290: "...analysis performed in the very resuscitation phase of sepsis could have...". I suggest to remove "sepsis" (since only 3 of 38 patients were septic) and change it for "shock".

5.- Conclusions: I suggest to emphasize that the observed hemodynamic incoherence was detected in an heterogeneous critically ill population.

Round 2

Reviewer 1 Report

The authors have revised their manuscript and the manuscript was significantly improved. However, there are still major concerns that need to be addressed:

Major:

The authors have confirmed, that this was a retrospective study. Moreover, the patients and indications for fluid challenge were highly heterogenous and the patient number was very small for a retrospective study. Moreover, a sample size calculation was not performed. Therefore, the results of the study should be interpreted very cautiously and at the most may indicate that certain effects on macro- and microcirculation after fluid challenge may be present but this cannot be confirmed by this study design and needs to be confirmed in further studies. Therefore, the discussion, conclusion and abstract should be adapted and definitive conclusions should be removed, instead, the results should be interpreted in the context of the study design.

For example, the reviewer suggests that the abstract conclusion should be adapted as the following:

"The results of this heterogenous population of critically ill patients might suggest incoherence in fluid responsiveness between systemic and microvascular hemodynamics although this has yet to be confirmed by prospective studies with adequate patient number based and a priori sample size calculations."

For the conclusion, the reviewer suggests that it should be adapted as the following:

"In this heterogeneous patients’ population of critically ill patients, this study may point to an incoherence between CI-responsiveness and PPV-responsiveness in almost half of fluid challenges although this has yet to be confirmed by prospective studies with adequate patient number and a priori sample size calculations. As such, the definition of fluid responsiveness may be challenged and may include, in some patients, microcirculatory assessment to evaluate the effects of fluid administration on tissue perfusion. Whether this approach is feasible and might be used to titrate fluid administration in critically ill patients, has to be demonstrated in larger studies. "

Moreover, since the measurements were performed directly after completion of fluid challenge, it is very much unclear, how macrohemodynamics and microcirculation would have been after equilibration of fluid effects and the results are limited to acute effects but cannot be used to assess the situation after equilibration. This should be mentioned as limitation.

Minor:

The reviewer suggests to present metric data as mean with confidence intervals, since the authors state that data was normally distributed instead of IQR.

Author Response

  1. The authors have confirmed, that this was a retrospective study. Moreover, the patients and indications for fluid challenge were highly heterogenous and the patient number was very small for a retrospective study. Moreover, a sample size calculation was not performed. Therefore, the results of the study should be interpreted very cautiously and at the most may indicate that certain effects on macro- and microcirculation after fluid challenge may be present but this cannot be confirmed by this study design and needs to be confirmed in further studies. Therefore, the discussion, conclusion and abstract should be adapted and definitive conclusions should be removed, instead, the results should be interpreted in the context of the study design. For example, the reviewer suggests that the abstract conclusion should be adapted as the following: "The results of this heterogenous population of critically ill patients might suggest incoherence in fluid responsiveness between systemic and microvascular hemodynamics although this has yet to be confirmed by prospective studies with adequate patient number based and a priori sample size calculations."

Authors’ response: The abstract has been adjusted according to this suggestion.

  1. For the conclusion, the reviewer suggests that it should be adapted as the following: "In this heterogeneous patients’ population of critically ill patients, this study may point to an incoherence between CI-responsiveness and PPV-responsiveness in almost half of fluid challenges although this has yet to be confirmed by prospective studies with adequate patient number and a priori sample size calculations. As such, the definition of fluid responsiveness may be challenged and may include, in some patients, microcirculatory assessment to evaluate the effects of fluid administration on tissue perfusion. Whether this approach is feasible and might be used to titrate fluid administration in critically ill patients, has to be demonstrated in larger studies. "

Authors’ response: The conclusions have been adjusted, accordingly.

  1. Moreover, since the measurements were performed directly after completion of fluid challenge, it is very much unclear, how macrohemodynamics and microcirculation would have been after equilibration of fluid effects and the results are limited to acute effects but cannot be used to assess the situation after equilibration. This should be mentioned as limitation.

Authors’ response: We have added this issue, accordingly.

  1. The reviewer suggests to present metric data as mean with confidence intervals, since the authors state that data was normally distributed instead of IQR.

Authors’ response: We think there has been a misunderstanding on this issue. Original text stated “Data were expressed as median and interquartile range (25th to 75th percentiles). The Kolmogorov-Smirnov test was used to verify the normality of distribution of continuous variables.” The sequence of sentences should have been inverted as first normality was tested therefore median and IQRs were used as data were not normally distributed. We have changed the text as such “Data were expressed as median and interquartile range (25th to 75th percentiles) as the Kolmogorov-Smirnov test showed a non-normal of distribution of continuous variables” – we hope to have clarified this issue.